# Exploring the Relationship between Career Satisfaction and University Learning Using Data Science Models

Sofía Ramos-Pulido [1,*] , Neil Hernández-Gress [1] and Gabriela Torres-Delgado [2]

1 School of Engineering and Science, Tecnologico de Monterrey, Monterrey 64849, Mexico; ngress@tec.mx
2 School of Humanities and Education, Tecnologico de Monterrey, Monterrey 64849, Mexico; gtorresd@tec.mx
* Correspondence: a00834077@tec.mx

**Abstract:** Current research on the career satisfaction of graduates limits educational institutions in devising methods to attain high career satisfaction. Thus, this study aims to use data science models to understand and predict career satisfaction based on information collected from surveys of university alumni. Five machine learning (ML) algorithms were used for data analysis, including the decision tree, random forest, gradient boosting, support vector machine, and neural network models. To achieve optimal prediction performance, we utilized the Bayesian optimization method to fine-tune the parameters of the five ML algorithms. The five ML models were compared with logistic and ordinal regression. Then, to extract the most important features of the best predictive model, we employed the SHapley Additive exPlanations (SHAP), a novel methodology for extracting the significant features in ML. The results indicated that gradient boosting is a marginally superior predictive model, with 2–3% higher accuracy and area under the receiver operating characteristic curve (AUC) compared to logistic and ordinal regression. Interestingly, concerning low career satisfaction, those with the worst scores for the phrase "how frequently applied knowledge, skills, or technological tools from the academic training" were less satisfied with their careers. To summarize, career satisfaction is related to academic training, alumni satisfaction, employment status, published articles or books, and other factors.

**Keywords:** graduates; educational institutions; data mining; career satisfaction; income satisfaction; life satisfaction; important features; gradient boosting; knowledge; skills; technological tools

## 1. Introduction

Career satisfaction is an important indicator of career success and is the most commonly cited subjective measure [1]. Although career satisfaction is closely related to job satisfaction, the concept of career satisfaction considers additional factors. Career satisfaction represents our feelings and emotions regarding our careers, i.e., whether we have achieved our aims or believe our future careers to be promising [1].

Career satisfaction is relevant for educational institutions, students, and graduates because it is related to alumni satisfaction [2], life satisfaction [3–5], and salary or income [4,6].

Refs. [4,6] determined the significance of salary for understanding career satisfaction. In particular, ref. [4] evaluated a positive correlation between current salary and career satisfaction, i.e., current salary positively predicts career satisfaction. Ref. [6] reported that 37-year-old individuals with higher incomes expressed greater career satisfaction. These findings suggest that an adequate salary contributes significantly to career satisfaction.

Building upon the insights from [4], who highlighted the significance of salary in understanding career satisfaction, it is worth noting that the existing relevant research literature states that the association between career and life satisfaction is related through time [5]. Additionally, ref. [3] conducted a meta-analysis and inferred that career satisfaction may be an important indicator of the quality of professional life.

In the target demographic of university graduates, previous studies primarily examined career satisfaction in terms of life or job satisfaction [3–5,7], salary [4,6,8], core self-evaluations [7,9,10], age [4,8], and cynicism [7,11]. In addition to studies reporting the positive impact of a person's goal orientation [9], training [12], and the impact of behavioral competencies on levels of career satisfaction [4], we did not discover other studies that highlighted the factors promoting career satisfaction of graduates.

This study adds to the literature by confirming the associations between career satisfaction and both salary and life satisfaction. We examine whether levels of career satisfaction can be predicted from life satisfaction and salaries and whether other features can provide better predictions of the levels of career satisfaction. Furthermore, this research aims to report general results or vital features distinguishing between graduates experiencing professional satisfaction and those lacking the same. We intend to assist educational institutions in devising approaches that will improve career satisfaction among future graduates.

Although regression models are widely applied for analyzing the career satisfaction of graduates, this study will demonstrate that alternative machine learning (ML) models can be implemented for accurately predicting career satisfaction. Owing to the quality and quantity of information and the type of applied data science models, this research hypothesizes that career satisfaction can be explained through the acquired survey variables. We also consider that certain characteristics are more important for determining career satisfaction among graduates and that data science models can outperform traditional regression models.

Unlike previous research, this study acquired data from a different educational institution and implemented data science models to predict career satisfaction among graduates. First, this study contributes to the existing literature on career satisfaction. Second, it contributes to the field of data science by exploring algorithms such as support vector machines, neural networks, and gradient boosting. Third, it contributes to the application of explainable artificial intelligence by identifying the essential features with SHapley Additive exPlanations (SHAP) values. Finally, this study is the first research to delve into the concept of career satisfaction based on several related factors, such as the frequency with which graduates apply knowledge, skills, or technological tools from academic training.

*Related Works*

A summary of recent studies analyzing career satisfaction is presented in Table 1, including the authors' names, year of publication, and the corresponding multivariate statistical model applied therein. For future comparisons with the ordinal model, these studies reported the following metrics of model performance: $R^2$ or adjusted $R^2$. Furthermore, these studies considered career satisfaction as the response variable. As observed from Table 1, the reference metric of $R^2 = 0.33$ reported by [9] will be used herein for comparison with the performance of the proposed model. Ref. [13] reported the positive impact of mentoring on career success during postgraduate specialist training based on a longitudinal investigation using hierarchical multiple regression analysis in a sample of 326 medical school graduates. The objective and subjective career success and career satisfaction were considered the response variables. In particular, for career satisfaction, the only statistically significant variable was a full-time job. As a recommendation, the study concluded that students must be advised to seek mentors.

Ref. [9] examined the impact of academic competencies and goal orientation on early career satisfaction. They used a sample of 247 alumni who graduated from a business school in the past five years. They applied a hierarchical regression model with interactions between the features. As reported, grades and academic competencies demonstrated limited influence in predicting early career success. Nevertheless, they reported that an individual's goal orientation is a significant determinant of career satisfaction. In the same study, they advised performing auxiliary activities to reveal the motivation for learning and developing competencies relevant to careers.

In his initial study conducted in 2012, ref. [7] aimed to shed light on the early professional experiences of recently graduated nurses during their first two years of practice. Additionally, the author sought to investigate the factors that could predict career satisfaction. The results revealed that workplace factors, which were amenable to change, had a substantial influence on both job and career satisfaction, as well as on the likelihood of new graduates leaving their positions.

**Table 1.** Related studies.

| Study | Year | Response Variable | Method | Metric |
|---|---|---|---|---|
| Stamm and Buddeberg-Fischer [13] | 2011 | Career satisfaction | HLR | $\nabla R^2 = 0.10$ |
| Laschinger [7] | 2012 | Career satisfaction | HLR | |
| van Dierendonck and van der Gaast [9] | 2013 | Career satisfaction | MLR | $R^2 = 0.33$ |
| Amdurer et al. [4] | 2014 | Career satisfaction | SEM | |
| Levy [14] | 2015 | Career satisfaction | HLR | $R^2 = 0.23$ |
| Kelly and Northrop [15] | 2015 | Career satisfaction | MLR | |
| Laschinger et al. [11] | 2016 | Career satisfaction | HLR | $R^2 = 0.21$ |
| Faupel-Badger et al. [8] | 2017 | Career satisfaction | MLR | |
| Erdogan et al. [16] | 2018 | Career satisfaction | SEM | $R^2 = 0.21$ |
| Holtschlag et al. [10] | 2019 | Career satisfaction | MLR | $R^2 = 0.27$ |
| Wojcik et al. [12] | 2020 | Career satisfaction | MLR | |
| Khalafallah et al. [17] | 2020 | Career satisfaction | MLR | |

HLR: hierarchical linear model; MLR: multiple linear regression; HRA: hierarchical regression analysis; SEM: structural equation modeling; MLR: multiple logistic regression; and MOR: multivariable ordinal regression.

Subsequently, in 2016, ref. [11] revisited the subject of the predictors of career satisfaction and turnover intentions, focusing primarily on a sample predominantly comprised of women (91.8%). Their study revealed that while situational and personal factors contributed significantly to the variations in career satisfaction and turnover intentions among new graduate nurses, cynicism and psychological capital were significant predictors of career satisfaction.

Meanwhile, the authors of [4] delved into the various competencies of emotional, social, and cognitive intelligence and their impact on career satisfaction. The study used a sample of 266 alumni from a university and found several statistically significant features for predicting career satisfaction, including current salary, age at graduation, adaptability, and GMAT percentile, among others. The findings of the study have significant implications for individuals and organizations seeking to enhance career success and satisfaction.

Considering only female graduates, ref. [14] employed a sample of 350 graduates aged between 30 and 60 years. They utilized hierarchical multiple regression models to evaluate the relationship between workaholism and career satisfaction. Intriguingly, their study revealed that among the various facets of workaholism, work enjoyment emerged as the sole component significantly linked to career satisfaction. This finding underscores the importance of fostering a positive and enjoyable work environment to enhance overall career satisfaction among this demographic.

In a longitudinal study conducted in [15], the researchers studied a group of full-time teachers who engaged in three different interventions. Notable among the statistically significant factors influencing career satisfaction were selectivity, gender, and Hispanic background. It was underscored that, for all teachers, irrespective of their initial selectivity, career satisfaction is an important predictor of attrition. Furthermore, the study revealed that male teachers reported lower levels of career satisfaction, whereas Hispanic teachers demonstrated less burnout and more career satisfaction compared to their peers.

Regarding the relationship between subjective assessments and career satisfaction, ref. [16] proposed a model to analyze the circumstances and reasons for the association of career satisfaction with perceived overqualification. They used a sample of 143 graduates and path analyses, determining that relative deprivation and perceived overqualification were negatively related to career satisfaction. Similarly, ref. [10] examined whether core

self-evaluations predict career satisfaction. They tested their hypothesis using a sample set of 139 alumni with an average work experience of seven years. They found that core self-evaluations and organizational embeddedness were positively associated with career satisfaction. Furthermore, positive emotions in relation to aims via interactions with organizational embeddedness served as an accurate predictor of career satisfaction beyond salary.

More recently, ref. [12] conducted a study involving a sample of 2050 army physicians, using logistic regression to identify that rank, training, and workplace exhibited statistical significance in explaining career satisfaction. In a similar vein, ref. [17] examined the impact of COVID-19 by incorporating career satisfaction as an output feature. Their findings indicated a paradoxical result, as they reported a moderate burnout rate alongside a remarkably high career satisfaction rate among neurosurgery residents. This conclusion suggests that, despite experiencing fatigue, these residents found great fulfillment in their roles, particularly in their capacity to assist.

## 2. Methods

### 2.1. Sample

The database used in this study is the property of Tecnológico de Monterrey University. Tecnológico de Monterrey, commonly known as Tec de Monterrey or ITESM (Instituto Tecnológico y de Estudios Superiores de Monterrey), is a prominent private, nonprofit university with multiple campuses in Mexico. In the latest regional rankings by the international evaluation firm Quacquarelli Symonds, Tec de Monterrey was positioned among the top four universities in Latin America and is the premier university in Mexico [18].

During the celebrations of its 75th anniversary in 2017, the university conducted a study to measure the economic and social impact on its graduates since its foundation in 1943. The survey invitation was electronically sent to all alumni, totaling 269,482. The survey results comprised 17,898 observations before data cleaning. The data treatment was rigorously anonymous; only one person could access the original database. The university provided the dataset without any personal information.

The dataset analyzed during the current study is not publicly available because it was used in the current study under a confidentiality agreement. The complete list of features used is provided in this section and the Appendix A. A synthetic dataset only will be provided if required. We confirm that informed consent was obtained from all subjects participating in the survey. No underage respondents responded to the survey.

The contents of this survey were validated through inter-judge agreement, which had previously validated the items' contents and forms. We confirm that a committee of research professors and several administrators from the university approved all protocols. The QS Intelligence Unit Team and analysts from the university conducted the descriptive analysis for this survey, and the university owns a report.

For this research, the dataset used was already cleaned, coded, and preprocessed. The research department consented to the use of the dataset, filtered by graduates who provided responses regarding the target variable: career satisfaction. The final dataset contained information on 12,180 graduates and 119 features.

In particular, 43% of the graduates were affiliated with the Engineering and Sciences School, followed by the Business School (37%), and the remaining 20% studied in either the Humanities and Education School; Medical School; School of Social Sciences and Government; or School of Architecture, Art, and Design. To summarize, a higher proportion of male graduates (60%) participated in the survey compared to female graduates (40%). The highest number of responses was recorded for those aged 30–39 years (38%), followed by 40–49 years (26%), <30 years (20%), 50–59 years (12%), and >60 years (4%). Note that 54% of graduates pursued a postgraduate degree. In terms of their current address at the time of the survey, a higher number of graduates resided in the region's center (41%), followed by the north (32%), and west and south (11%), whereas the remaining graduates resided abroad (16%).

## 2.2. Dependent Variable

Previous studies have implemented similar approaches to evaluating career satisfaction. In this regard, the survey and analysis reported in [19] used a five-point Likert scale and inquired "Thinking very generally about your satisfaction with your overall career in medicine currently". Similarly, the surveys analyzed in [12,13,20] used ten-, seven-, and five-point Likert scales for a specific question: "how satisfied are you with your career?".

Similar to the existing literature, this study measured **career satisfaction** using the level of agreement with the statement, "I am completely satisfied with my professional career". Responses were based on a seven-point Likert scale (1: strongly disagree; 7: strongly agree). In terms of career satisfaction, the highest number of responses was recorded at level 6 (32.6%), followed by level 5 (25.3%), level 7 (24.4%), level 4 (10%), and less than 4 (7.7%). To summarize, approximately 80% of the surveyed graduates were satisfied or highly satisfied with their current careers (levels 5–7).

## 2.3. Independent Variables

The input features employed in this study were nominal, dichotomous, ordinal, and numerical. The descriptions of the input features used in this study are presented in Table 2 and Table A1. The descriptions of the five essential features are detailed below.

**Table 2.** Feature description.

| Feature Name | Type | Levels |
|---|---|---|
| Sex | Dichotomous | 1: Woman; 0: Man |
| Age | Numerical | 23–71 years |
| Scholarship percentage | Numerical | 1%–100% |
| Postgraduate degree | Dichotomous | 1: Yes; 0: Not |
| Study abroad | Dichotomous | 1: Yes; 0: Not |
| Weekly working hours | Numerical | 0–60 h |
| Tenure in previous job | Numerical | 0–50 years |
| Years working abroad | Numerical | 0–10 years |
| Academic training | Numerical | 0–10 |
| Academic training comparison | Ordinal | Less: 1.0; Equal: 2.0; More: 3.0 |
| Years working as a general director | Numerical | 0–11 |
| Years working as a subdirectory | Numerical | 0–11 |
| Number of personnel in charge | Ordinal | 0: zero employees; 1 : between 1 and 10 employees; 2 : between 11 and 20; 3 : between 31 and 40; 4 : more than 40 |
| Organization size in first and current job | Ordinal | 0: zero employees; 1 : between 1 and 10 employees; 2 : between 11 and 50 employees; 3 : between 51 and 100 employees; 4 : more than 100 employees |
| First and current salary | Ordinal | Level 1; Level 2; Level 3; Level 4 |

*Income satisfaction* was measured based on the level of agreement with the statement, "I am completely satisfied with my income". The responses were recorded on a seven-point Likert scale.

*Life satisfaction* was measured based on the level of agreement with the statement, "I am completely satisfied with my life". The responses were recorded on a seven-point Likert scale (1: strongly disagree; 7: strongly agree).

*I have achieved what I consider important* was measured based on the level of agreement with the statement, "So far, I have achieved the most important things I want in my life". The responses were recorded on a seven-point Likert scale (1: strongly disagree; 7: strongly agree).

*I would not change anything* was measured according to the level of agreement with the statement, "If I had to live my life over again, I would not change a thing". The responses were recorded on a seven-point Likert scale (1: strongly disagree; 7: completely agree).

*The frequency of applying knowledge* was measured based on the question "how often do you apply the specific knowledge, skills, and/or technological tools of your academic program in your current occupation?" The responses were recorded as numbers ranging from 0 to 10.

### 2.4. Statistical Analysis and Imbalance Classification Problem Treatment

This research used Chi-squared tests to derive the associations between career satisfaction and the other features of the dataset. Ordinal and logistic regression was applied as a baseline model to compare the predictive power of the machine learning models. Moreover, the numerical values of the $R^2$ metric of the ordinal regression models were used to confirm the results against similar research reports. If the ordinal regression passes the assumption of proportional odds assumption, we will compare the results of this statistical procedure and the best machine learning model. The proportional odds assumption for ordinal regression was assessed with the Brant test using the 'Brant' package in R [21]. The Statsmodels library [22] in Python was used to implement chi-squared tests and fit the ordinal regression with a 'logit' distribution.

Furthermore, a class imbalance problem was addressed to treat the imbalanced distribution of the target variable. Class imbalance problems occur when certain classes contain a greater number of individuals than others. Consequently, the class imbalance problem has emerged as one of the greatest issues in data mining [23]. The algorithms appropriately classified the majority class but neglected the minority classes. As per [24,25], heuristics is generally used to address the class imbalance problem, focusing on balancing data such as oversampling and undersampling. For instance, in the oversampling technique, data can be balanced by replicating the minority class samples. Although undersampling involves reducing the sizes of the majority classes, it is limited because it loses valuable information [23]. However, it should be noted that the majority of research on class imbalance pertains to two-class imbalance problems [26], and only a few methods can efficiently manage multiclass or ordinal class imbalances [24,26].

In this study, for career satisfaction, classes ranging from 1 to 4 were selected less frequently than classes 5–7. Here, sampling techniques were implemented to balance the classes of the target features. However, these techniques did not improve the predictive performance of the models across these seven categories to the desired extent. As this study primarily aimed to derive insights into low and high satisfaction levels, classes 1–4 were merged into a single category. Consequently, the lowest scores of various metrics improved to 70%. Therefore, the target variable at this stage included three levels: level 1 with a point scale ranging from 1 to 4, level 2 with points 5 and 6, and level 3 with a point scale of 7. As per this categorization, 17.7% of the participants indicated low satisfaction, 57.9% indicated moderate satisfaction, and 24.4% reported high satisfaction.

### 2.5. Supervised Learning Models and Bayesian Optimization

The machine learning methods employed to assess career satisfaction included the decision tree, random forest, gradient boosting, support vector machine, and neural network models. The scikit-learn library [27] was used to implement the machine learning models and the metrics for evaluating the models' performance.

The performance of machine learning models often depends on selecting the optimal hyperparameter configuration, a task that often demands extensive expertise in ML algorithms and the application of suitable hyperparameter optimization techniques, as highlighted in [28]. Currently, Bayesian optimization has been highlighted in [29] and others as a powerful and popular technique used in machine learning for the hyperparameter tuning problem. Bayesian optimization constructs a probabilistic model of a given function and leverages this model to make decisions about where to evaluate the function next [30]. This

innovative approach eliminates the need for exhaustive searches, such as grid or random searches, which involve evaluating many hyperparameter configurations.

For this study, we used the Bayes opt library [31] in Python to tune the hyperparameters of all the implemented data science models. The hyperparameters evaluated were those that are typically tuned according to [28]. Table 3 summarizes the hyperparameters tuned.

**Table 3.** Hyperparameters tuned.

| SVM | NNK | GB |
|---|---|---|
| C: $[1\times10^3, 4\times10^4]$<br>Kernel: Poly and rbf<br>Degree: [1,4]<br>Gamma: $[1\times10^{-6}, 2]$ | Activation: logistic, tanh, ReLU<br>Layers: [30,500]<br>Neurons: [1,10]<br>Alpha: $[1\times10^{-7}, 1\times10^{-1}]$<br>Learning rate: $[1\times10^{-8}, 1\times10^{-1}]$<br>Optimizer: Adam | Max. depth: [1,10]<br>Number of estimators: [50,140] |
| **DT** | **RF** | **LR** |
| Max. depth: [1,10]<br>Criterion: gini and entropy | Max. depth: [1,10]<br>Number of estimators: [50,140] | C: $[0, 4\times10^4]$<br>Penalty: l1 and l2 |

SVM: support vector machine; NNK: neural network; GB: gradient boosting; DT: decision tree; RF: random forest; LR: logistic regression.

Model performance was assessed through a random cross-validation (CV) approach. This involved creating five random splits, allocating 80% of the data to the training set and 20% to the testing set for each split. In each split, data mining algorithms and logistic and ordinal regression were trained on the training set, and their predictive performance was evaluated using the corresponding test set. The overall performance of the algorithms was then estimated by computing the average value across the five splits.

The most appropriate hyperparameter values were determined using a 5-fold cross-validation (CV) approach for each model and random CV split. The training set was divided into five folds, with the models being trained on each hyperparameter value using four of these folds. The resulting model was then validated on the remaining portion of the data, and this process was iterated five times, covering each subsample in turn. The validation metric for each hyperparameter element represents the average performance over these five repetitions. The "optimal" hyperparameter values were identified as those within the grid that produced the highest value for the validation metric. Finally, these optimal values were employed to make predictions on the testing set. Accuracy, recall, precision, F1, and AUC (area under the ROC curve) were assessed to evaluate the performance of the models.

While accuracy stands out as a direct and versatile metric applicable to a variety of classification problems, its reliability diminishes in the context of imbalanced datasets. In scenarios where one class dominates the dataset, achieving high accuracy becomes possible by simply predicting the majority class. On the contrary, although the interpretation of the AUC may not be as immediate as that of accuracy, the AUC demonstrates greater resilience in the face of imbalanced datasets. This becomes particularly pivotal when confronted with situations where one class significantly outnumbers the other. The AUC evaluates the model across all classification thresholds, and as indicated in [32], the AUC score represents the probability that the classifier will rank a randomly chosen positive instance higher than a randomly chosen negative instance, offering a comprehensive measure of overall performance. Consequently, given the imbalance in the target variable analyzed in this study, a more robust evaluation is achieved by considering additional metrics such as the AUC, recall, precision, and F1.

### 2.6. Feature Importance

The most important features for the best predictive model were extracted with SHapley Additive exPlanations (SHAP) using the SHAP Python library [33]. Initial insights were obtained using the feature importance plot, employing SHAP values specifically tailored for

tree models and ensembles of trees [34]. This plot effectively illustrates the general impact of features on predicting the target variable. Although it identifies the most crucial features, discerning the direction of their effects poses a challenge. To address this, the analysis incorporated the SHAP summary plot [34]. Organized in descending order of significance on the y-axis and utilizing color-coded dots on the x-axis to represent the SHAP values (ranging from blue for low values to red for high values), this plot not only gauges the importance of features but also elucidates the direction and magnitude of their effects.

To provide a more granular understanding of individual predictions, a waterfall plot was employed [33]. Beginning at the model's expected output, each row of the waterfall plot delineates how each feature's positive (red) or negative (blue) contribution influences the transition from the expected model output over the background dataset to the model output for the specific prediction [33]. This comprehensive suite of SHAP analyses ensures a robust exploration of feature importance, directionality, and individual prediction explanations for the enhanced interpretability of the predictive model.

## 3. Results

### 3.1. Statistical Analysis

In the results of the $\chi^2$ tests, many values of $p_{value}$ were approximated to zero (see Table 4), providing robust evidence against the independence hypothesis. The factors of life and career satisfaction showed a strong statistically significant association. Comparably, income satisfaction and the response to the phrase "I have achieved what I consider important" were strongly associated. The results of the $\chi^2$ test ($\chi^2$, $p_{value} \approx 0$) supported this significant association and indicated that the response to the phrase "I would not change anything" was significantly associated with career satisfaction $\chi^2$, $p_{value} \approx 0$).

The correlations between career, life, and income satisfaction were significant and positive, as shown in Figure 1. The correlation between career satisfaction and gender was negligible, as was the correlation with the number of working hours a week and publishing books or articles. According to [35], negligible correlations correspond to the range of 0 to $\pm 0.2$, weak correlations correspond to the range of $\pm 0.2$ to $\pm 0.4$, moderate correlations correspond to the range of $\pm 0.41$ to $\pm 0.6$, and strong correlations correspond to the range of $\pm 0.6$ to $\pm 0.8$. Interesting associations and weak correlations were observed between career satisfaction and alumni satisfaction, as measured by the questions, "I would study again at my university" and "I am committed to supporting my university".

Despite the weak correlation, the Chi-square test between career satisfaction and the frequency of applying knowledge, skills, and technological tools yielded statistical significance. Two notable trends can be seen in Figure 2. When graduates perceive that they use their knowledge, skills, or technological tools less, the percentage of low job satisfaction increases. Conversely, as they consider applying more of these resources, the percentage of high professional satisfaction increases.

Another significant trend was observed between career satisfaction and academic training. In Figure 3, it is clear that when academic training evaluations are low, career satisfaction is also low. Similarly, when evaluations are higher, the percentage of moderate career satisfaction is considerably higher compared to low and high levels.

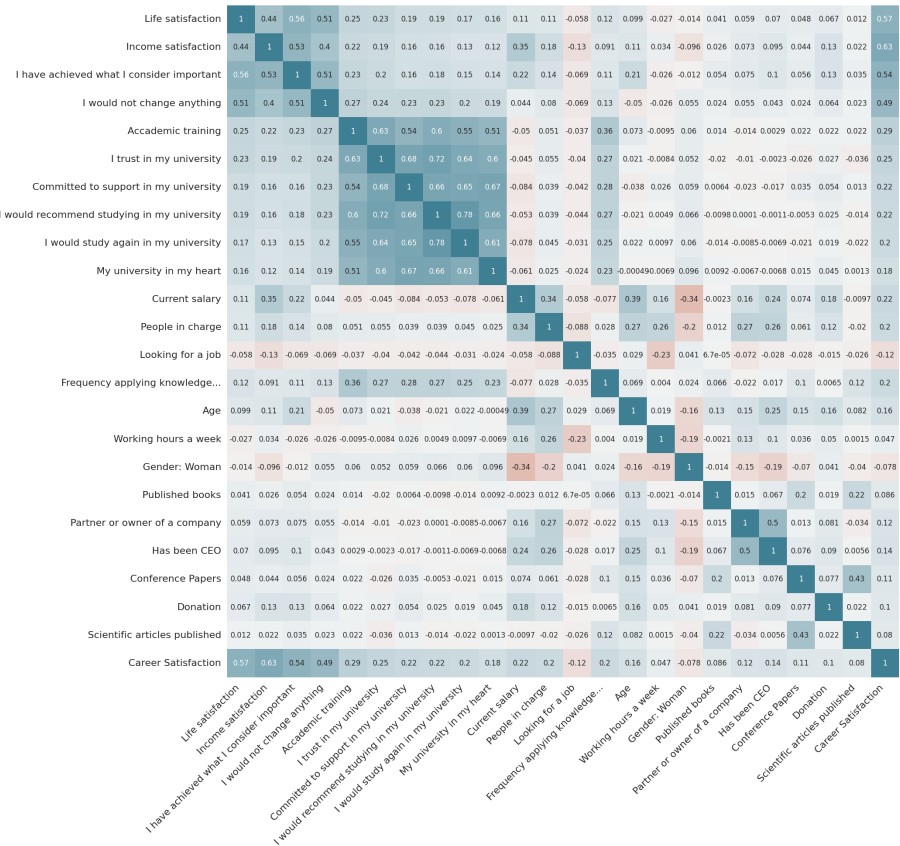

**Figure 1.** Correlations. Solid blue indicates a strong positive correlation, while lighter shades indicate negligible positive correlations. The same applies to negative correlations, represented by the color orange.

**Table 4.** $\chi^2$ tests.

| | Career Satisfaction | |
|---|---|---|
| | $\chi^2$ **Test** | $p_{value}$ |
| Life satisfaction | 10,865.4 | 0.00 |
| Income satisfaction | 10,586.3 | 0.00 |
| I have achieved what I consider important | 8067.66 | 0.00 |
| I would not change anything | 5244.12 | 0.00 |
| Academic training | 1761.19 | 0.00 |
| I trust my university | 1588.55 | $1.66 \times 10^{-292}$ |
| I am committed to supporting my university | 1208.09 | $2.46 \times 10^{-213}$ |
| I recommend studying at my university | 1116.88 | $1.63 \times 10^{-194}$ |
| I would study again at my university | 993.90 | $2.81 \times 10^{-169}$ |
| My university is in my heart | 903.00 | $9.64 \times 10^{-151}$ |
| Current salary | 755.53 | $9.14 \times 10^{-149}$ |
| Frequency of applying knowledge | 817.24 | $7.16 \times 10^{-137}$ |
| People in charge | 596.03 | $1.97 \times 10^{-106}$ |
| Looking for a job | 413.07 | $4.33 \times 10^{-86}$ |
| I have been a general director | 236.43 | $3.25 \times 10^{-48}$ |
| I have been a partner or owner of a company | 195.87 | $1.44 \times 10^{-39}$ |
| Age | 810.49 | $1.15 \times 10^{-31}$ |
| Conference papers | 141.81 | $4.16 \times 10^{-28}$ |
| Working hours | 954.81 | $1.78 \times 10^{-25}$ |
| Make donations | 128.97 | $2.12 \times 10^{-25}$ |
| Gender | 101.78 | $1.07 \times 10^{-19}$ |
| I have published books | 96.87 | $1.13 \times 10^{-18}$ |
| I have published scientific articles | 90.96 | $1.91 \times 10^{-17}$ |

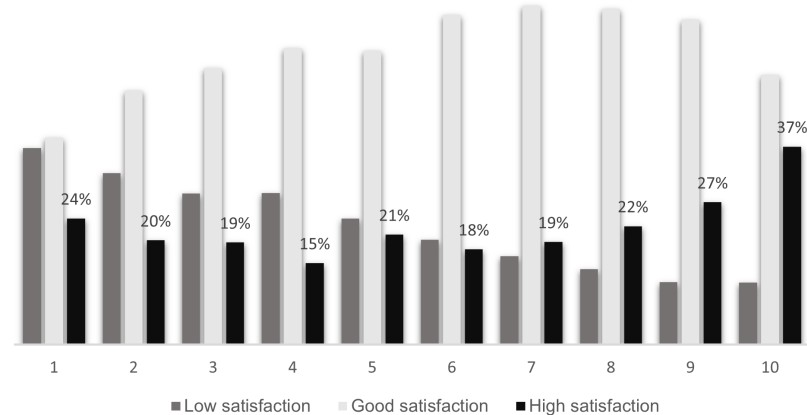

**Figure 2.** Career satisfaction vs. frequency.

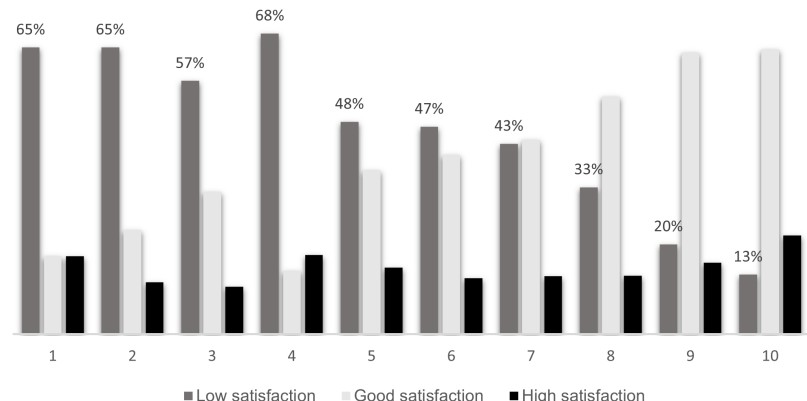

**Figure 3.** Career satisfaction vs. academic training.

### 3.2. Supervised Learning Models

The hyperparameters for each model were determined through Bayesian optimization after five repetitions of five-fold cross-validation. The selected criteria were entropy and a maximum depth of 6 for the decision tree model; a maximum depth of 3 and 110 estimators for the gradient boosting model; a maximum depth of 9 and 130 estimators for the random forest model; with a regularization parameter (C) of 28,664 and an L2 penalty for the logistic regression model; a polynomial kernel, degree 1, and C of 13,217 for the support vector machine model; and finally, a rectified linear unit (ReLU) activation function, a single layer with 100 units, an alpha of $1 \times 10^{-4}$, and a learning rate of $1 \times 10^{-3}$ for the neural network model.

The models' performance is summarized in Table 5 using various metrics. The worst accuracy and AUC were observed for the decision tree model, with values of 70% and 0.83, respectively. In comparison, the random forest, logistic regression, and ordinal regression models all exhibited an accuracy value of 71% and AUC values of 0.86, 0.85, and 0.85, respectively. As observed, the gradient boosting model achieved a superior AUC, and the random forest algorithm delivered higher precision. Ultimately, the best recall, accuracy, and F1 values were obtained with the gradient boosting, SVM, and neural network models. In contrast, the worst recall and F1 values were obtained with the random forest and logistic regression models, with overall deviations between 1 and 3 points.

**Table 5.** Results for the metrics.

|  |  | SVM | NNK | DT | GB | RF | LR | OLR |
|---|---|---|---|---|---|---|---|---|
|  | Accuracy | **74** | **74** | 70 | **74** | 71 | 71 | 71 |
|  | AUC | 85 | 86 | 83 | **87** | 86 | 85 | 85 |
| Precision | Macro-average | 73 | 73 | 71 | 75 | **79** | 73 | 71 |
|  | Weighted average | 74 | 73 | 71 | 74 | **75** | 72 | 71 |
| Recall | Macro-average | **66** | **66** | 62 | **66** | 58 | 61 | 63 |
|  | Weighted average | **73** | **73** | 70 | **73** | 71 | 71 | 71 |
| F1 | Macro-average | **69** | **69** | 64 | **69** | 61 | 64 | 67 |
|  | Weighted average | **73** | **73** | 69 | **73** | 68 | 70 | 70 |

SVM: support vector machine; NNK: neural network; DT: decision tree; GB: gradient boosting; RF: random forest; LR: logistic regression; OLR: ordinal logistic regression. AUC: area under the ROC curve or area under the receiver operating characteristic curve.

The pseudo-R-squared score for the ordinal regression model was 0.34 in the training set over five repetitions. This value surpassed the R-squared value observed in most studies in the literature review section. Given this specific application, where ordinal regression exhibited comparatively lower predictive power compared to the SVM, neural network, and gradient boosting models, it is evident that these models stand out as competitive and relevant approaches for evaluating the career satisfaction of graduates.

Regrettably, the assumptions of the proportional odds in ordinal regression were violated, as indicated by the brand test. Consequently, we could not employ ANOVA to identify statistically significant features explaining career satisfaction. For reference, Table A2 in the Appendix A displays the ANOVA results of the ordinal regression model.

Overall, gradient boosting was selected as the superior model because of its high performance across most metrics. Accordingly, in the following section, the most essential features were extracted using gradient boosting.

### 3.3. Important Features of Gradient Boosting

The first five significant features for predicting career satisfaction shown in Figure 4 were substantially beyond the discriminative power of the remaining features.

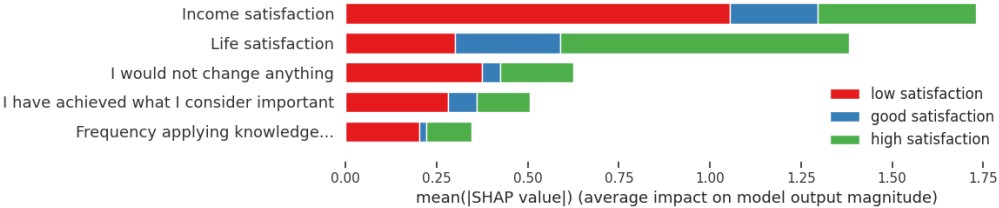

**Figure 4.** Feature importance with SHAP.

The high scores in life satisfaction, income satisfaction, "I would not change anything", "I have achieved what I consider important", "years working as a general director", "published books", and "published scientific articles" predicted high career satisfaction (Figure 5).

Low career satisfaction can be explained in order of significance by low scores in income satisfaction, "I would not change anything", "I have achieved what I consider important", life satisfaction, "frequency of applying knowledge, skills, and technological tools acquired from the academic program", and not having individuals in charge (Figure 6). Interestingly, graduates actively seeking jobs were predicted to experience low career satisfaction.

Figure 7 depicts an individual prediction associated with high career satisfaction. Interestingly, the high scores for the frequency of applying knowledge, skills, and technological tools acquired from the academic program increased this person's predicted probability of having high career satisfaction. Except for seniority, the high scores in the other features reported in this figure, for example, life and income satisfaction and academic training, also increased the predicted probability of having high career satisfaction.

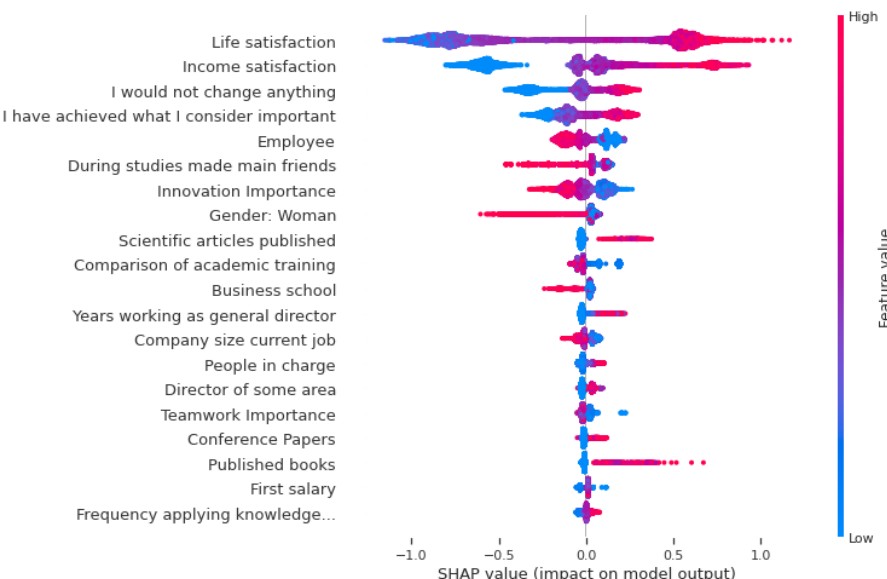

**Figure 5.** Important features for high career satisfaction.

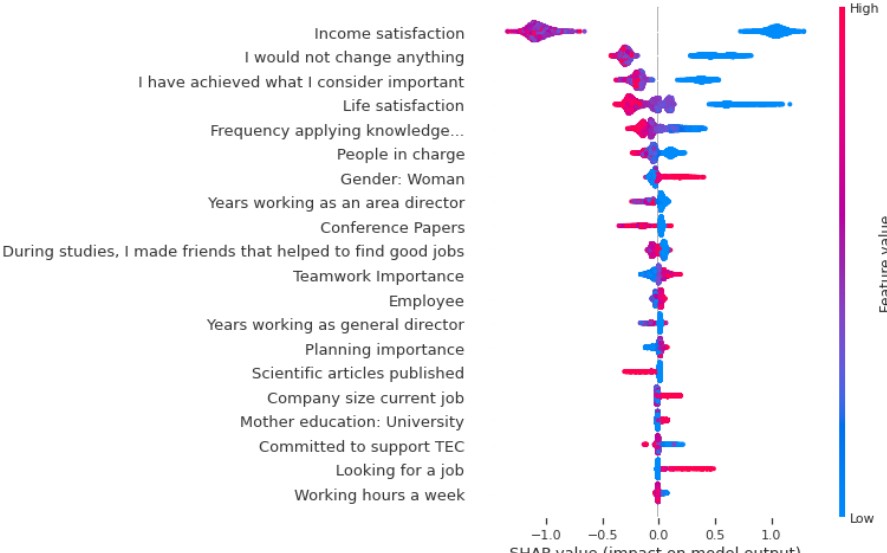

**Figure 6.** Important features for low career satisfaction.

Similarly, Figure 8 depicts an individual prediction of low career satisfaction. Interestingly, looking for a job increased this person's predicted probability of having low satisfaction with his or her career. Also, low scores in, for example, income satisfaction increased the possibility of experiencing low satisfaction with a career. For the score of 8 for the frequency of applying knowledge, skills, and technological tools acquired from the academic program and the score of 1 for life satisfaction, the negative values, $-0.26$ in both cases, implied probabilities of less than 0.5 for experiencing low career satisfaction. In other words, this combination of life satisfaction and applied knowledge is typically not associated with low career satisfaction.

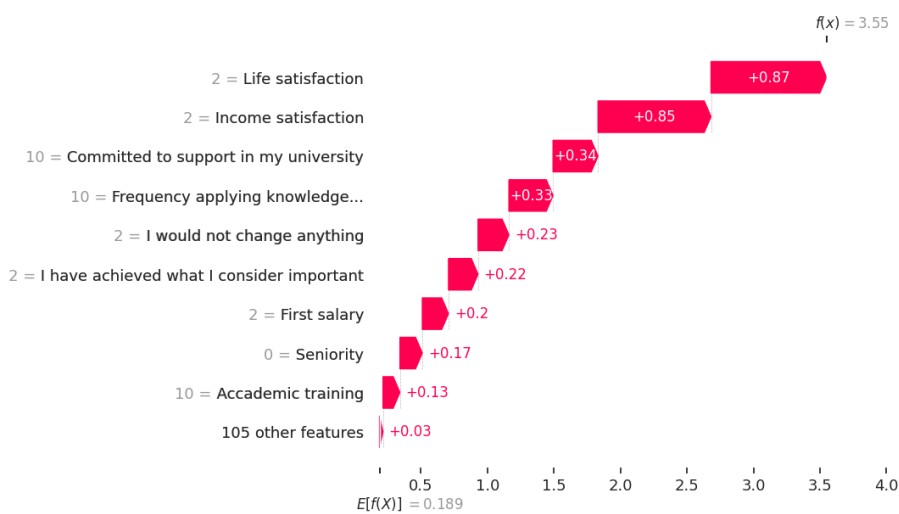

**Figure 7.** Individual prediction of high career satisfaction.

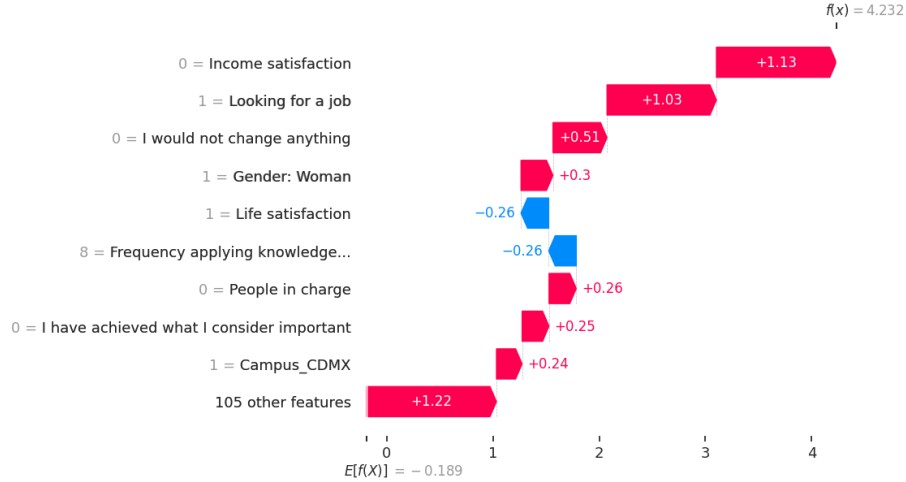

**Figure 8.** Individual prediction of low career satisfaction.

## 4. Discussions and Conclusions

This is the first research to inform educational institutions that the degree to which a graduate applies the skills or technological tools learned in their curriculum impacts their career satisfaction. Graduates who do not apply the knowledge acquired during their academic training are frequently dissatisfied with their careers. Similarly, in the evaluation of academic training, those considered to have undergone adequate academic training consider themselves between satisfied and highly satisfied, whereas graduates identifying their academic training as the worst are generally less satisfied with their careers.

In the search for the best predictive model to classify the levels of the target variable, gradient boosting demonstrated the best predictive performance. This is in line with expectations because the existing literature on machine learning classifies the gradient boosting algorithm as a competitive supervised learning procedure [36,37]. The gradient boosting model outperformed the decision tree, logistic regression, and ordinal regression models across all metrics; the SVM and neural network models in AUC and precision; and the random forest model in most metrics, except for precision.

The gradient boosting model was predictively competitive compared to traditional models such as hierarchical regression analysis because the ordinal regression model, which exhibited lower performance compared to the gradient boosting model, achieved a pseudo-R-squared metric of 0.34, which was almost equal to the $R^2 = 0.33$ of the hierarchical model proposed in [13].

Regarding other practical implications, the results of this cross-sectional study confirm the associations reported in [3,5] between career satisfaction and life satisfaction. To summarize, life satisfaction and income are the most crucial features for predicting high career satisfaction. However, income satisfaction is the most important feature for predicting a low level of career satisfaction.

Furthermore, this study confirmed the association between salary and career satisfaction reported in [4,6], and the association between high age and high career satisfaction reported in [38,39], where current salary and age were highly associated with career satisfaction. In this study, 91% of graduates older than 60 years were satisfied or highly satisfied with their careers.

Regarding employability, in this investigation, the outcomes indicated that current employment status (measured from employed "yes" or "no") and career satisfaction are associated, i.e., unemployed graduates seeking a job are generally less satisfied with their careers. In terms of the level of employment, similar to the study conducted in [8], where the authors cited that greater leadership roles are associated with higher career satisfaction, the results in this study revealed that the number of years working as a general director is linked to increased career satisfaction.

In addition to the associations confirmed between career satisfaction and other factors studied in the literature, this study highlighted the relationship between high career satisfaction and research activities (e.g., publishing articles and books). Moreover, our findings revealed statistically significant relationships between career satisfaction and graduates' commitment to supporting the university, as well as their inclination to recommend the university. These insights have positive implications for universities, underscoring the potential benefits derived from fostering a supportive and satisfying academic environment.

Based on these findings, some practical implications and suggestions for educational institutions to improve the career satisfaction of graduates could include the following:

- Create courses and study plans that emphasize the real-world application of the knowledge and skills students acquire in the classroom.
- Provide practical opportunities like internships that simulate actual workplace scenarios to prepare students for their future careers [40].
- Build solid connections with businesses to know what skills are needed in the job market.
- Ask for opinions from both graduates and employers to find out where the curriculum can be enhanced [41].
- Inspire current students by showcasing stories of graduates who have successfully used what they learned in their careers [42].
- Encourage students to have a mindset of ongoing learning and adaptability, and teach them to view challenges as chances to develop and improve.

We conclude by stating that this research demonstrated how current alternative data science procedures outperform traditional regression methodologies in prediction and offer thorough procedures like SHAP to extract the features with the best predictive power for the target variables. Several factors are associated with career satisfaction, including actionable features that educational institutions could use to improve career satisfaction among graduates.

## 5. Limitations

This study is limited because our conclusions were based on the responses and self-evaluations of the graduates. The results represent graduates from only one educational institution and thus additional research is required involving graduates from other universities worldwide. Furthermore, more features should be used in the explanatory and predictive models. Thus, future research should examine other actionable features students can develop to improve career satisfaction.

**Author Contributions:** Data collection, N.H.-G.; Conceptualization, S.R.-P., N.H.-G. and G.T.-D.; Methodology, S.R.-P. and N.H.-G.; Experimentation, analysis, and interpretation, S.R.-P.; Visualization, S.R.-P.; Writing—original draft, S.R.-P. and N.H.-G.; Writing—review and editing, S.R.-P., N.H.-G. and G.T.-D. All authors have read and agreed to the published version of the manuscript

**Funding:** This research received no external funding.

**Institutional Review Board Statement:** The survey was carried out following the relevant guidelines and regulations. We confirm that a committee of research professors and several administrators from the university approved all protocols.

**Informed Consent Statement:** We confirm that informed consent was obtained from all subjects participating in the survey. No underage respondents responded to the survey.

**Data Availability Statement:** The dataset analyzed during the current study is not publicly available because it was used in the current study under a confidentiality agreement. The complete list of features used is provided in Section 2 and the Appendix A synthetic dataset only will be provided upon reasonable request to the corresponding author.

**Acknowledgments:** The authors thank Tecnologico de Monterrey University for allowing them to use the database of graduates developed during the 75th anniversary celebrations. The authors would also like to thank Tecnológico de Monterrey University and Conacyt for supporting S.R.-P. with a Ph.D. scholarship.

**Conflicts of Interest:** The authors declare no conflicts of interest.

## Abbreviations

The following abbreviations are used in this manuscript:

| | |
|---|---|
| DT | Decision trees |
| GB | Gradient boosting |
| RF | Random forest |
| LR | Logistic regression |
| OLR | Ordinal logistic regression |
| SVM | Support vector machine |
| NNK | Neural network |
| SHAP | SHapley Additive exPlanations |
| AUC | Area under the ROC curve |
| CV | Cross-validation |
| ROC | Receiver operating characteristic |

## Appendix A

*Appendix A.1*

**Table A1.** Description of features.

| Feature Name | Type | Category Names |
|---|---|---|
| Years working as a CEO | Numerical | 0–11 |
| Years working in the government | Numerical | 0–11 |
| Donations | Dichotomous | 1: Yes; 0: No |
| Business administration councils | Dichotomous | 1: Yes; 0: No |
| Nonprofit organizations | Dichotomous | 1: Yes; 0: No |
| Voluntary work | Dichotomous | 1: Yes; 0: No |
| Num. businesses, funded | Numerical | 0–11 |
| Num. businesses, working | Numerical | 0–11 |
| Innovation importance | Numerical | 0–6 |
| Communication importance | Numerical | 0–6 |
| Teamwork importance | Numerical | 0–6 |
| Negotiation importance | Numerical | 0–6 |
| Planning importance | Numerical | 0–6 |
| Negotiation importance | Numerical | 0–6 |
| I trust my university | Numerical | 0–10 |
| I am committed to my university | Numerical | 0–10 |

**Table A1.** *Cont.*

| Feature Name | Type | Category Names |
|---|---|---|
| I would study again at university | Numerical | 0–10 |
| I recommend my university | Numerical | 0–10 |
| My university is in my heart | Numerical | 0–10 |
| Parents' occupation | Nominal | Business owner, employee, freelancer, manager, public server, housewife, and other |
| Parents' education | Nominal | Without a college degree, technical career, primary school, secondary school, high school, bachelor's degree, and postgraduate degree. |
| Studies | Dichotomous | Master's, doctorate, specialties, and medical specialty residency |
| Current and birth address region | Nominal | Center, foreign, north, south, and west |
| School | Nominal | Business, engineering, and other |
| Campus | Nominal | CDMX, center, MTY, north, south, online, and west |
| Sector, first and current job | Nominal | Primary, secondary, quaternary, tertiary, and other |
| Employment situation | Dichotomous | Paid employee, partner or business owner, independent professional, looking for a job, do not want a job, and student. |
| Management experience | Dichotomous | Subdirector, CEO, and director of area |
| Government employment | Dichotomous | "Director of an institute, agency, or social department", or "deputy, senator, or governor" |
| Publications | Dichotomous | Books, chapter books, research articles, articles in opinion magazines, and conference papers |
| Inventions | Dichotomous | Process innovation and product innovation |
| Productions | Dichotomous | Software, movies, advertisements, artistic works, architectural designs, and musical compositions |
| At university, you met someone who is or was | Dichotomous | A sentimental couple, a partner in companies or organizations, main friends, and someone who made it easy for you to find a good job |

*Appendix A.2*

**Table A2.** Statistically significant features with ordinal regression.

| | Target Variable: Career Satisfaction | | | |
|---|---|---|---|---|
| | Coef. | Std. Err. | z | $p_{value}$ |
| 1. Income satisfaction | 111.87 | 3.163 | 35.373 | $4.46 \times 10^{-274}$ |
| 2. Life satisfaction | 92.81 | 3.121 | 29.74 | $2.34 \times 10^{-194}$ |
| 3. I would not change anything | 32.56 | 2.428 | 13.41 | $5.36 \times 10^{-41}$ |
| 4. I have achieved what I consider important | 34.23 | 2.77 | 12.38 | $3.41 \times 10^{-35}$ |
| 5. Frequency of applying knowledge | 5.18 | 0.70 | 7.442 | $9.92 \times 10^{-14}$ |
| 6. Innovation importance | −9.87 | 1.28 | −5.88 | $1.03 \times 10^{-13}$ |
| 7. Teamwork importance | −10.46 | 1.42 | −7.37 | $1.67 \times 10^{-13}$ |
| 8. Looking for a job | −89.25 | 14.10 | −6.33 | $2.46 \times 10^{-10}$ |
| 9. Negotiation importance | −8.08 | 1.31 | −6.18 | $6.60 \times 10^{-10}$ |
| 10. Planning importance | −7.50 | 1.28 | −5.88 | $4.00 \times 10^{-09}$ |
| 11. Communication importance | −8.19 | 1.41 | −5.83 | $5.69 \times 10^{-9}$ |
| 12. Employee | −13.51 | 3.57 | −3.78 | $4.04 \times 10^{-8}$ |
| 13. Father's education: postgraduate | −32.38 | 7.85 | −4.12 | $3.72 \times 10^{-5}$ |
| 14. Academic training comparison | −15.37 | 3.79 | −4.06 | $5.01 \times 10^{-5}$ |
| 15. Independent professional | −32.94 | 8.37 | −3.94 | $8.27 \times 10^{-5}$ |
| 16. People in charge | 5.38 | 1.37 | 3.93 | $8.58 \times 10^{-5}$ |
| 17. Father's education: university | −27.94 | 7.28 | −3.84 | $1.24 \times 10^{-4}$ |
| 18. Gender | −13.51 | 3.57 | −3.79 | $1.53 \times 10^{-4}$ |
| 19. Father's education: high school | −31.59 | 8.47 | −3.73 | $1.92 \times 10^{-4}$ |
| 20. Current address: north | −33.56 | 9.21 | −3.64 | $2.69 \times 10^{-4}$ |
| 21. Academic training | 6.14 | 1.70 | 3.60 | $3.13 \times 10^{-4}$ |
| 22. Business school | −16.45 | 4.74 | −3.47 | $5.16 \times 10^{-4}$ |
| 23. Published books | 29.66 | 8.59 | 3.45 | $5.54 \times 10^{-4}$ |
| 24. Current address: foreign | −35.46 | 10.29 | −3.45 | $5.71 \times 10^{-4}$ |
| 25. Father's education: secondary | −31.00 | 9.38 | −3.30 | $9.55 \times 10^{-4}$ |
| 26. Nonprofit organizations | 20.16 | 6.23 | 3.24 | $1.21 \times 10^{-3}$ |
| 27. Father's education: technical career | −27.72 | 8.76 | −3.17 | $1.55 \times 10^{-3}$ |
| 28. Current address: center | −26.79 | 8.90 | −3.01 | $2.61 \times 10^{-3}$ |
| 29. Musical compositions | 31.30 | 10.49 | 2.98 | $2.84 \times 10^{-3}$ |

**Table A2.** *Cont.*

| | Target Variable: Career Satisfaction | | | |
|---|---|---|---|---|
| | Coef. | Std. Err. | z | $p_{value}$ |
| 30. Owner of a business | −24.36 | 8.17 | −2.98 | $2.85 \times 10^{-3}$ |
| 31. Published scientific articles | 20.17 | 6.79 | 2.97 | $2.96 \times 10^{-3}$ |
| 32. Mother education: High school | −46.85 | 16.06 | −2.92 | $3.52 \times 10^{-3}$ |
| 33. Conference paper published | 14.32 | 5.16 | 2.77 | $5.54 \times 10^{-3}$ |
| 34. Mother's education: University | −44.18 | 15.94 | −2.77 | $5.58 \times 10^{-3}$ |
| 35. During studies, made main friends | 8.94 | 3.26 | 2.74 | $6.15 \times 10^{-3}$ |
| 36. Company size, current job | −4.90 | 1.79 | −2.74 | $6.21 \times 10^{-3}$ |
| 37. Mother's education: technical career | −40.32 | 15.93 | −2.53 | $1.14 \times 10^{-2}$ |
| 38. Committed to supporting my university | 2.65 | 1.07 | 2.47 | $1.34 \times 10^{-2}$ |
| 39. Current salary | 5.38 | 2.25 | 2.40 | $1.66 \times 10^{-2}$ |
| 40. Mother's education: secondary | −38.89 | 16.47 | −2.36 | $1.82 \times 10^{-2}$ |
| 41. Work sector, first job: primary | −28.00 | 11.86 | −2.36 | $1.82 \times 10^{-2}$ |
| 42. Student | 38.32 | 16.30 | 2.35 | $1.87 \times 10^{-2}$ |
| 43. I do not have and do not want a job | −37.35 | 16.10 | −2.32 | $2.03 \times 10^{-2}$ |
| 44. Mother's education: Postgrad. | −38.18 | 16.57 | −2.30 | $2.12 \times 10^{-2}$ |
| 45. Birth address: center | −16.27 | 7.28 | −2.23 | $2.55 \times 10^{-2}$ |
| 46. Work sector, first job: tertiary | −9.62 | 4.36 | −2.21 | $2.73 \times 10^{-2}$ |
| 47. Current address: west | −23.16 | 11.27 | −2.05 | $3.99 \times 10^{-2}$ |
| 48. Mother's education: primary | −33.85 | 16.58 | −2.04 | $4.12 \times 10^{-2}$ |
| 49. Birth address: west | −21.32 | 10.51 | −2.03 | $4.25 \times 10^{-2}$ |

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
