# Peer review of "Exploring the Relationship between Career Satisfaction and University Learning Using Data Science Models"

_informatics, doi:10.3390/informatics11010006_

Round 1

Reviewer 1 Report

Comments and Suggestions for Authors

Overall, this study presents a novel approach to addressing an important issue. However, there are some concerns that could be addressed to enhance the clarity, reliability, and applicability of the findings.

  1. It is important to discuss potential biases and limitations in the dataset to understand the generalizability of the results.
  2. The article mentions using SHAP for feature extraction, but it would be helpful to have more information on how these features were interpreted and their implications for educational institutions. It would be great if the authors could provide actionable strategies based on the identified features.
  3. A more comprehensive discussion on the strengths and limitations of the chosen evaluation metrics, namely accuracy and AUC, would add methodological robustness to the study and research findings.
  4. It would be great if the authors could address ethical considerations for the ethical integrity of this study, given the nature of predicting personal outcomes based on survey responses.
Comments on the Quality of English Language

The writing of this article is well-structured and understandable for readers.

Author Response

Dear revisor

I appreciate your time and the valuable suggestions for enhancing the manuscript's quality. Please find the responses to your questions and comments below.

Thank you.

Reviewer 2 Report

Comments and Suggestions for Authors

This paper will be of interest to readers in the fields of education, data science, and career development. The study utilizes data science models to predict career satisfaction based on survey responses from university alumni. However, it would be beneficial if the authors provided more context for the study. What is the nature of the participating university? Why did the authors choose this specific university? Is it ranked higher in the country? The introduction could benefit from more explicit statements regarding the significance and novelty of the study.

The research design is appropriate, focusing on using data science models. The methods section is adequately described, especially concerning the sample size. However, concerning the survey, what is the reliability and validity of the survey? Did the authors develop the survey? How? Are there any sub-constructs in this survey? Why did the authors analyze each item instead of sub-constructs?

The results are presented clearly, and the choice of gradient boosting as the superior model is well-justified. Regarding the implications of the study, it would be great if the authors suggested practical implications for educational institutions or other stakeholders.

Author Response

Dear Reviewer

Thank you for dedicating your time to reviewing the manuscript and providing valuable suggestions to improve its quality. Below, you will find the responses to your questions and comments.

Best regards
